# ACCELERATING MULTI-BLOCK CONSTRAINED OPTIMIZATION THROUGH LEARNING TO OPTIMIZE

## ABSTRACT

Learning to Optimize (L2O) approaches, including algorithm unrolling, plug-and-play methods, and hyperparameter learning, have garnered significant attention and have been successfully applied to the Alternating Direction Method of Multipliers (ADMM) and its variants. However, the natural extension of L2O to multi-block ADMM-type methods remains largely unexplored. Such an extension is critical, as multi-block methods leverage the separable structure of optimization problems, offering substantial reductions in per-iteration complexity. Given that classical multi-block ADMM does not guarantee convergence, the Majorized Proximal Augmented Lagrangian Method (MPALM), which shares a similar form with multi-block ADMM and ensures convergence, is more suitable in this setting. Despite its theoretical advantages, MPALM's performance is highly sensitive to the choice of penalty parameters. To address this limitation, we propose a novel L2O approach that adaptively selects this hyperparameter using supervised learning. We demonstrate the versatility and effectiveness of our method by applying it to the Lasso problem and the optimal transport problem. Our numerical results show that the proposed framework outperforms popular alternatives. Given its applicability to generic linearly constrained composite optimization problems, this work opens the door to a wide range of potential real-world applications.

## 1 INTRODUCTION

Optimization is a fundamental and essential process in machine learning and data science (Bottou et al., 2018). It involves fine-tuning algorithms, models, or systems to make them perform at their best on complex tasks. The goal of optimization is to find the best solution from a set of possible choices (i.e., feasible set), often with the aim of minimizing or maximizing a particular function (i.e., loss function). In the context of machine learning, optimization often refers to the process of adjusting a model's parameters to minimize the error or loss function. This is typically achieved through various classical algorithmic frameworks such as gradient descent (GD) (Cauchy et al., 1847), stochastic gradient descent (SGD) (Robbins and Monro, 1951), and more advanced methods like Adam (Kingma and Ba, 2014) and RMSprop (Hinton et al., 2012), to mention just a few. Machine learning approaches can play significant roles in designing efficient optimization algorithms and greatly improving the performance of well-designed algorithms. This field of research is known as Learning to Optimize (L2O) which attracts growing attention (Chen et al., 2022). In particular, given a class of optimization problems, machine learning approaches can be used to effectively predict the performance of different hyperparameters and configurations of an optimization algorithm, thereby guiding the search for optimal configurations and even automating the process of tuning (Hutter et al., 2011; Feurer and Hutter, 2019).

### 1.1 OPTIMIZATION MODEL

Following the research scheme of L2O, we consider leveraging machine learning techniques in solving the following class of generic multi-block convex composite optimization problems:

$$\min_{y_i \in \mathbb{Y}_i,\ i=1,\ldots,p} f_\xi(y_1, \ldots, y_p) + g(y_1) \quad \text{s.t.} \quad \sum_{i=1}^{p} \mathcal{A}_i^* y_i = c, \qquad (P(\xi))$$

where $\mathbb{Y}_i$ is finite dimensional Euclidean space for $i = 1, \ldots, p$, $g : \mathbb{Y}_1 \rightarrow (-\infty, +\infty]$ is a closed proper convex (possibly nonsmooth) function, $f_\xi : \mathbb{Y}_1 \times \cdots \times \mathbb{Y}_p \rightarrow (-\infty, +\infty)$ is a continuously differentiable convex function that depends on a random variable $\xi \in \Xi$ sampled from a fixed distribution $\mathcal{P}$, $c \in \mathbb{X}$ is a given vector in the finite dimensional Euclidean space $\mathbb{X}$, and $\mathcal{A}_i : \mathbb{X} \rightarrow \mathbb{Y}_i$ is a linear mapping for $i = 1, \ldots, p$. For notational simplicity, we denote $y := (y_1; \ldots; y_p)^T \in \mathbb{Y} := \mathbb{Y}_1 \times \cdots \times \mathbb{Y}_p$. Moreover, we define the linear mapping $\mathcal{A} : \mathbb{X} \rightarrow \mathbb{Y}$ as $\mathcal{A} := (\mathcal{A}_1; \ldots; \mathcal{A}_p)$. Then, we see that $\sum_{i=1}^p \mathcal{A}_i^T y_i = \mathcal{A}^* y$, for all $y \in \mathbb{Y}$. In this paper, we assume that the following mild assumption with respect to the function $f_\xi$ always holds.

**Assumption 1 (Majorization of $f_\xi$)** *There exists a fixed self-adjoint positive semidefinite linear operator $\Sigma : \mathbb{Y} \rightarrow \mathbb{Y}$ such that, for any $\xi \in \Xi$,*

$$f_\xi(y) \leq q_\xi(y; y') := f_\xi(y') + \langle \nabla f_\xi(y'), y - y' \rangle + \frac{1}{2} \|y - y'\|_\Sigma^2, \quad \forall\, y, y' \in \mathbb{Y}.$$

For later usage, we shall partition $\Sigma$ into $p \times p$ sub-blocks as $\Sigma = [\Sigma_{ij}]_{1 \leq i,j \leq p}$, where $\Sigma_{ij} : \mathbb{Y}_j \rightarrow \mathbb{Y}_i$, $1 \leq i, j \leq p$, are linear mappings. We can see that Assumption 1 is indeed quite mild. For example, if $f_\xi$ has a Lipschitz continuous gradient for every $\xi \in \Xi$ with a uniformly bounded Lipschitz constant, then Assumption 1 holds. In the latter case, $\Sigma$ can be chosen as a diagonal matrix. Here, we allow non-diagonal $\Sigma$ in order to obtain a more accurate majorization of $f_\xi$ when defining the majorized augmented Lagrangian function.

An optimization problem of the form $(P(\xi))$ is of great interest, mainly due to its excellent modeling power. Indeed, many optimization problems from practical applications in the fields of statistical and machine learning, engineering, and image and signal processing can be formulated as instances of $(P(\xi))$. These applications include composite convex quadratic conic programming (Li et al., 2018b; Liang et al., 2022), penalized and constrained regression (James et al., 2019), compressed sensing and sparse coding (Tibshirani, 1996; Chen et al., 2001; Donoho, 2006; Li et al., 2018a), matrix and tensor completion (Candes and Plan, 2010; Liu et al., 2012), regularized optimal transport (Peyré et al., 2019; Yang et al., 2024), and consensus optimization and federated learning (Boyd et al., 2011; Zhang et al., 2021). Note that for these modern applications in the era of big data, the number of blocks $p$ and the dimension of a $\mathbb{Y}_i$ can be large. Moreover, in practice, one usually needs to solve a sequence of optimization problems of the same form $(P(\xi))$ as the random variable $\xi$ being sampled from the distribution $\mathcal{P}$. Hence, solving these optimization problems efficiently at scale is essential for practical considerations.

As a convex optimization problem, $(P(\xi))$ can be solved via applying existing algorithms designed for convex optimization. High-order methods, including interior point methods (IPMs) (Nesterov and Nemirovskii, 1994) and augmented Lagrangian methods (ALMs) (Hestenes, 1969; Fiacco and McCormick, 1990; Rockafellar, 1976), are commonly adopted due to their fast convergence rates and reliability in computing highly accurate solutions. Typically, at each iteration of a high-order method, one needs to solve a linear system or a convex (composite) quadratic programming subproblem, leading to excessive computational time per iteration. This can make the algorithm not scalable for large-scale problems. To design scalable algorithmic frameworks, recent years have witnessed significant advancements in developing and analyzing first-order methods (FOMs) for solving $(P(\xi))$. Given the presence of a potentially nonsmooth regularization term and linear constraints, prevalent first-order methods in machine learning, such as GD, SGD, Adam, and RMSprop, cannot be directly applied to the interested model $(P(\xi))$.

One of the most preferred approaches is the two-block alternating direction method of multipliers (ADMM) (Gabay and Mercier, 1976), which is a robust and scalable first-order method used for solving large-scale linearly constrained convex optimization problems; see Appendix A for more details. Though convergent under mild conditions, the two-block ADMM views $y = (y_1, \ldots, y_p)$ as a single block, ignoring potential separable structures which may be beneficial to explore. Moreover, solving the corresponding subproblem at each iteration involving the whole $y$ can be time-consuming, leading to high iteration complexity of the two-block ADMM. This motivates the direct extension of the two-block ADMM to the multi-block ADMM which favors the separable structure of $y$ in the objective and constraints; see Appendix B. At each iteration of the multi-block ADMM, $p$ subproblems of smaller sizes are solved individually, following a Jacobi-type updating rule. This results in reduced per-iteration complexity. However, the convergence of the above direct extension is no longer guaranteed (Chen et al., 2016). To resolve this issue, a multi-block ADMM-type method

known as the Majorized Proximal Augmented Lagrangian Method (MPALM) was proposed recently by Chen et al. (2021). The key ingredient of MPALM is to replace the Jacobi-type updating rule with a symmetric Gauss-Seidel-type updating rule, ensuring elegant convergence properties, under mild conditions.

## 1.2 Our contributions

The practical performance of the MPALM can be sensitive to the value of the augmented Lagrangian penalty parameter. Hence, to get satisfactory numerical performance, one needs to adaptively adjust its value. However, the adjustment can be highly problem dependent and requires domain expertise (Lam et al., 2018). Given the assumption that $\xi \in \Xi$ is sample from a fixed distribution $\mathcal{P}$, developing data-driven frameworks for selecting the parameter is of significant interest to yield excellent empirical performance and enhance the applicability and efficiency of ADMM-type algorithms for solving challenging optimization problems of the form $(P(\xi))$. This paper aims at integrating data-driven machine learning techniques into the MPALM for solving the class of optimization problems $(P(\xi))$. Specifically, the contributions of this paper are summarized as follows.

- We propose a simple yet effective L2O framework for learning the hyperparameter of a majorized proximal augmented Lagrangian method, a convergent multi-block ADMM-type method for solving the challenging optimization problems in the form of $(P(\xi))$. Our work continues the research theme in L2O and enhances the applicability of machine learning techniques in designing efficient optimization algorithms for generic constrained optimization, which is not fully explored in the literature.

- Numerically, we consider two important practical applications, the Lasso problem and the discrete optimal transport problem, and showcase that the proposed framework is highly effective via numerical experiments and comparisons with state-of-the-art approaches. Given the excellent modeling power of $(P(\xi))$, our work further motivates various potential real-world applications from other domains.

**Notation**. The following notation will be used throughout this paper. Let $\mathbb{E}$ be a finite dimensional Euclidean space, the standard inner product is denoted as $\langle \cdot, \cdot \rangle$ and the associated induced norm is denoted as $\|\cdot\|$. Let $S : \mathbb{E} \to \mathbb{E}$ be a self-adjoint positive semidefinite linear operator, the weighted-norm associated with $S$ is denoted as $\| \cdot \|_S$, i.e., for any $x \in \mathbb{E}$, $\|x\|_S = \sqrt{\langle x, Sx \rangle}$. For a differential function $f : \mathbb{E} \to \mathbb{R}$, its gradient is denoted as $\nabla f$. If $f : \mathbb{E} \to \mathbb{R} \cup \{\pm\infty\}$ is an extended-valued convex function, then the effective domain of $f$ is denoted as $\mathrm{dom}(f) := \{x \ : \ f(x) < \infty\}$. Let $x \in \mathrm{dom}(f)$, then the subgradient of the convex function $f$ at $x$ is denoted as $\partial f(x) := \{v \ : \ f(x') \geq f(x) + \langle v, x' - x \rangle, \ \forall x'\}$.

## 2 Related work

**Learn to optimize (L2O).** L2O is a modern and effective approach towards designing optimization algorithms that reach a new level of efficiency. A comprehensive survey of L2O can be found in (Chen et al., 2022). Our work is closely related to different subjects of L2O, including the configuration and hyperparameter learning (Hutter et al., 2011; Feurer and Hutter, 2019), plug-and-play approaches (Venkatakrishnan et al., 2013), and algorithm unrolling (Monga et al., 2021). Particularly, a series of works targeting unrolling algorithmic frameworks for the Lasso model (Tibshirani, 1996) (which as well as its dual problem are special cases of $(P(\xi))$) has attracted increasing attention (Gregor and LeCun, 2010; Moreau and Bruna, 2017; Perdios et al., 2017; Zhou et al., 2018; Ablin et al., 2019; Cowen et al., 2019; Hara et al., 2019; Liu and Chen, 2019). The fundamental algorithmic framework in these works is the Iterative Shrinkage Thresholding Algorithm (ISTA) (Beck and Teboulle, 2009). Despite being characterized by hyperparameter learning, our approach can also be viewed as the application of the algorithm unrolling to the MPALM. In this way, our work continues this line of research and provides an alternative L2O approach for efficiently solving challenging optimization problems from practical applications, including Lasso-type problems.

**ADMM-type methods.** Incorporating machine learning techniques with ADMM-type methods has drawn growing attention in recent years. The two-block ADMM was employed as the fundamental algorithmic framework in image denoising, in which the involved proximal operators are replaced

with learned operators (Venkatakrishnan et al., 2013; Brifman et al., 2016; Chan et al., 2016; Ryu et al., 2019). Great empirical success and comprehensive theoretical guarantees have been achieved in these works, making the plug-and-play ADMM approach one of the most popular and effective algorithms for image science. Just like unrolling the ISTA, unrolling the two-block ADMM and its linearized variant (also known as the primal dual hybrid gradient method (Chambolle and Pock, 2011)) has also attracted much attention; see e.g., (Sun et al., 2016; Rick Chang et al., 2017; Adler and Öktem, 2018; Cowen et al., 2019; Cheng et al., 2019; Xie et al., 2019) and references therein. The unrolled two-block ADMM-type methods have also been demonstrated to have excellent empirical performance. There are other machine learning techniques that are helpful for improving the practical performance of the two-block ADMM. For instance, (Zeng et al., 2022) successfully applied a reinforcement learning approach for selecting the hyperparameters in the two-block ADMM for distributed optimal power flow. While fruitful results have been established for L2O with two-block ADMM-type methods, results on combining L2O with multi-block ADMM-type methods, including MPALM, remain limited. We demonstrate in the present paper that the MPALM-based L2O approach is also effective when compared with existing state-of-the-art L2O approaches.

**Optimal transport (OT).** Recent years have seen a blossoming of interest in developing efficient solution methods for OT, which have numerous important applications, due partly to the essential metric property of its optimal solution (Cuturi, 2013z; Peyré et al., 2019). To solve OT problems efficiently at scale, the most popular first-order method is perhaps Sinkhorn's algorithm (Cuturi, 2013z). Sinkhorn's algorithm is a simple iterative method for finding optimal solutions of entropy-regularized OT problems. Thus, it only provides approximate solutions to the original OT problems. To get high quality approximations, one needs to choose a small entropy regularization parameter, which causes the convergence of Sinkhorn's algorithm to be extremely slow. Moreover, a small entropy regularization parameter may also result in numerical issues, making Sinkhorn's algorithm unstable. Hence, developing effective algorithmic frameworks for solving OT problems remains an active research direction (Genevay et al., 2016; Makkuva et al., 2020; Korotin et al., 2022; Liu et al., 2022; Chu et al., 2023; Hou et al., 2023; Zanetti and Gondzio, 2023; Yang et al., 2024). However, works focusing on L2O approaches for OT remain limited, to the best of our knowledge. Our work provides a feasible approach for solving OT problems reliably via combining L2O with MPALM.

## 3  THE MAJORIZED PROXIMAL AUGMENTED LAGRANGIAN METHOD

In this section, we provide a detailed introduction of the main algorithmic framework proposed in (Chen et al., 2021), namely the majorized proximal augmented Lagrangian method (MPALM), and explain how to cleverly choose the proximal term for the augmented Lagrangian function such that the resulting proximal ALM subproblem can be solved efficiently and analytically. The key idea of designing the proximal term is motivated by the Gauss-Seidel iterative method for solving symmetric positive definite linear systems.

Since the problem $(P(\xi))$ is a constrained convex optimization problem, we see that the first-order optimality conditions (also called the Karush-Kuhn-Tucker conditions) (Rockafellar, 1974) for problem $(P(\xi))$ are given by

$$0 \in \partial g(y_1) + \nabla f_\xi(y) + \mathcal{A}x, \quad \mathcal{A}^*y - c = 0. \tag{1}$$

For any point $(x^*, y^*) \in \mathbb{X} \times \mathbb{Y}$ that satisfies the above first-order optimality conditions, one can show that $y^*$ is a solution to problem $(P(\xi))$ and $x^*$ is the corresponding Lagrange multiplier (also known as the dual optimal solution). Assuming the solvability of the first-order optimality conditions (1), one may apply the classic augmented Lagrangian method for solving $(P(\xi))$. To this end, given a penalty parameter $\sigma > 0$, we define the augmented Lagrangian function as

$$\mathcal{L}_\sigma(y; x) := f_\xi(y) + g(y_1) + \langle \mathcal{A}^*y - c, x \rangle + \frac{\sigma}{2} \|\mathcal{A}^*y - c\|^2, \quad \forall (y, x) \in \mathbb{Y} \times \mathbb{X}.$$

Then given an initial point $x^0 \in \mathbb{X}$ and an increasing sequence of penalty parameter $\{\sigma_k\}$, the classical augmented Lagrangian iteratively performs the following updating scheme:

$$\begin{cases} y^{k+1} := \text{argmin} \left\{ \mathcal{L}_{\sigma_k}(y; x^k) \ : \ y \in \mathbb{Y} \right\}, \\ x^{k+1} := x^k + \sigma_k \left( \mathcal{A}^*y^{k+1} - c \right), \end{cases}$$

where $k \geq 0$ denotes the iteration counter. Though admitting excellent convergence properties (Rockafellar, 1976), the classical augmented Lagrangian method faces certain challenges: (1) $y^{k+1}$

with high accuracy can be difficult to obtain since one needs to solve a (possibly nonsmooth) convex optimization problem. (2) The potential separable structure in $y$ is not explicitly explored. We next show how to address these challenges via replacing the augmented Lagrangian function $\mathcal{L}_\sigma$ in the classical augmented Lagrangian method with a convex function (as the sum of a convex quadratic function and $g$) such that the resulting subproblems in updating $y$ are much easier to solve via fully exploring the potential separable structure in $y$.

Under Assumption 1, for a given $\xi \sim \mathcal{P}$ and the augmented Lagrangian penalty parameter $\sigma > 0$, we can define the majorized augmented Lagrangian function (with a slight abuse of the notation $\mathcal{L}$)

$$\mathcal{L}_{\xi,\sigma}(y; x, y') := q_\xi(y; y') + g(y_1) + \langle \mathcal{A}^* y - c, x \rangle + \frac{\sigma}{2} \left\| \mathcal{A}^* y - c \right\|^2, \quad \forall\, (y, x, y') \in \mathbb{Y} \times \mathbb{X} \times \mathbb{Y}.$$

Using the majorized augmented Lagrangian function, the majorized proximal augmented Lagrangian method (MPALM) for solving problem $(P(\xi))$ is presented in Algorithm 1; see Theorem 1 in Appendix C for the global convergence of Algorithm 1 under mild conditions. Note that under additional conditions, including a certain error-bound condition, Algorithm 1 can further be shown to have a local Q-linear convergence rate (Chen et al., 2021). Since these additional conditions are generally not easy to verify, we decide not to present the result here for simplicity. However, such a linear convergence is empirically observed based on numerical experience.

---

**Algorithm 1:** The majorized proximal ALM (MPALM).

---

**Input:** A fixed point $\xi \in \Xi$, an initial point $(x^0, y^0) \in \mathbb{X} \times \mathbb{Y}$, a self-adjoint linear mapping
$\quad\quad \mathcal{S} : \mathbb{Y} \to \mathbb{Y}$, the penalty parameter $\sigma > 0$, the step size $\tau \in (0, 2)$, and the maximum
$\quad\quad$ number of iterations $K > 0$.

1 **for** $k = 0, \ldots, K - 1$ **do**

2 $\quad\quad y^{k+1} = \operatorname{argmin} \left\{ \phi_{\xi,k}(y) := \mathcal{L}_{\xi,\sigma}(y; x^k, y^k) + \frac{1}{2} \left\| y - y^k \right\|_{\mathcal{S}}^2 \ : \ y \in \mathbb{Y} \right\}.$

3 $\quad\quad x^{k+1} = x^k + \tau\sigma \left( \mathcal{A}^* y^{k+1} - c \right).$

4 **end**

**Output:** $x^K$.

---

Given the excellent convergence properties of the MPALM, the next essential task is to choose the linear mapping $\mathcal{S} : \mathbb{Y} \to \mathbb{Y}$ such that the subproblems can be solved efficiently via exploring the separable structure associated with $y$. Note that the objective function for the optimization subproblem in MPALM, i.e., $\phi_{\xi,k}(\cdot)$, is a quadratic function of the form $\frac{1}{2} \langle x, (\mathcal{Q} + \mathcal{S})x \rangle + \langle c_k, x \rangle + d_k$, where $\mathcal{Q} := \sigma \mathcal{A} \mathcal{A}^* + \Sigma$ is a self-adjoint positive semidifinite linear operator, and $c_k, \ d_k$ depends on the iteration counter $k$; see (5). Suppose that $\mathcal{Q} = [\mathcal{Q}]_{ij}, 1 \le i, j \le p$, is partitioned into $p \times p$ blocks, and it is written as $\mathcal{Q} = \mathcal{U} + \mathcal{D} + \mathcal{U}^*$ where $\mathcal{D} := \operatorname{Diag}(\mathcal{Q}_{11}, \ldots, \mathcal{Q}_{pp})$ denotes the diagonal blocks of $\mathcal{Q}$ and $\mathcal{U}$ is the upper triangle part of $\mathcal{Q}$ such that $\mathcal{U}_{ij} = \mathcal{Q}_{ij}$ for $1 \le i < j \le p$ and $\mathcal{U}_{ij} = 0$ otherwise. (Li et al., 2019) provides an elegant approach for choosing $\mathcal{S}$ based on the idea of solving symmetric positive definite linear systems by Gauss-Seidel iterative method. In particular, assuming that $\mathcal{D}$ is nonsingular, the operator $\mathcal{S}$ is then choosen as $\mathcal{S} = \mathcal{U} \mathcal{D}^{-1} \mathcal{U}^*$, which is known as the SGS-operator of $\mathcal{Q}$. Then, one can show that $y^{k+1}$ can be computed exactly as follows:

$$\begin{aligned}
\tilde{y}_i^k &= \operatorname{argmin} \left\{ \mathcal{L}_{\xi,\sigma}(y_{<i}^k, y_i, \tilde{y}_{>i}^k; x^k, y^k) \ : \ y_i \in \mathbb{Y}_i \right\}, \quad i = p, \ldots, 2, \\
y_i^{k+1} &= \operatorname{argmin} \left\{ \mathcal{L}_{\xi,\sigma}(y_{<i}^{k+1}, y_i, \tilde{y}_{>i}^k; x^k, y^k) \ : \ y_i \in \mathbb{Y}_i \right\}, \quad i = 1, \ldots, p.
\end{aligned} \tag{2}$$

We refer the reader to Appendix C for the general case when $\mathcal{D}$ is not necessarily singular.

Applying (2), one immediately obtains a symmetric Gauss-Seidel-based majorized proximal ALM (see Algorithm 2). Consequently, we obtain an algorithm that takes a similar form of the multi-block ADMM with provable convergence. This is an appealing feature for practical applications since it allows one updating one block of the decision variables while keeping the remaining blocks fixed; see, e.g., the applications considered in Section 4. One can also observe that for $i = 2, \ldots p$, solving the associated subproblems involves only solving linear systems, which can be done nearly exactly via elementary linear algebra routines. For $i = 1$, the computation of $y_1$ is the proximal mapping of $g$. It is well-known that many important functions $g$ admit analytical proximal mapping or can be approximate efficiently and accurately (Parikh et al., 2014). Thus, the associated ALM subproblem can also be solved in low computational costs.

## 4 HYPERPARAMETER LEARNING

Though Algorithm 1 has attractive convergence properties, its practical performance is sensitive to the choice of the penalty parameter $\sigma$, based on our numerical experience. Practitioners typically adjust $\sigma$ dynamically in order to obtain better numerical performance (Lam et al., 2018). From a high level point of view, the following algorithm with adaptive penalty parameters, i.e., Algorithm 2, is commonly adopted in practice.

---

**Algorithm 2:** The symmetric Gauss-Seidel-based majorized proximal ALM with adaptive penalty parameters.

---

**Input:** A fixed point $\xi \in \Xi$, an initial point $(x^0, y^0) \in \mathbb{X} \times \mathbb{Y}$, the step size $\tau \in (0, 2)$, the maximum number of iterations $K > 0$, a positive integer $K_0 \leq K$, and the set of penalty parameters $\{\sigma_j \ : \ 0 \leq j \leq \lfloor K/K_0 \rfloor + 1\}$.

**1** **for** $k = 0, \dots, K - 1$ **do**
**2** $\quad$ Find $j$ such that $k \in [jK_0, (j+1)K_0)$ and set $\sigma = \sigma_j$.
**3** $\quad$ **for** $i = p, \dots, 2$ **do**
**4** $\quad\quad$ $\tilde{y}_i^k = \arg\min \left\{ \mathcal{L}_{\xi,\sigma}(y_{<i}^k, y_i, \tilde{y}_{>i}^k; x^k, y^k) \ : \ y_i \in \mathbb{Y}_i \right\}$.
**5** $\quad$ **end**
**6** $\quad$ **for** $i = 1, \dots, p$ **do**
**7** $\quad\quad$ $y_i^{k+1} = \arg\min \left\{ \mathcal{L}_{\xi,\sigma}(y_{<i}^{k+1}, y_i, \tilde{y}_{>i}^k; x^k, y^k) \ : \ y_i \in \mathbb{Y}_i \right\}$.
**8** $\quad$ **end**
**9** $\quad$ $x^{k+1} = x^k + \tau\sigma \left( \mathcal{A}^* y^{k+1} - c \right)$.
**10** **end**
**Output:** $x^K$.

---

However, the criterion guiding the adjustment of the penalty parameters can be highly heuristic, which depends on the problems being solved and often requires advanced domain knowledge. To alleviate this difficulty, we propose to apply the data-driven supervised learning approach for learning the penalty parameters $\{\sigma_j\}_{j=1}^{J}$ with $J := \lfloor K/K_0 \rfloor + 1$ such that the resulted algorithm performs empirically well when the random variable $\xi$ is sampled from a fixed distribution $\mathcal{P}$. In particular, for a fixed initial point $(x^0, y^0) \in \mathbb{X} \times \mathbb{Y}$ and a fixed step size $\tau \in (0, 2)$, we see that the output of Algorithm 2 depends only on the random variable $\xi \in \Xi$ and the penalty parameters $\{\sigma_j\}$. To emphasize this dependency, we shall denote the output of Algorithm 2 as $x^K(\xi, \{\sigma_j\})$, with slightly abuse of notation. Similarly, we denote $x^*(\xi)$ to be the optimal solution of problem $(P(\xi))$ depending on $\xi \in \Xi$. In order to find a good strategy of selecting the values of the penalty parameters $\{\sigma_j\}$, we consider solving the following empirical risk minimization problem:

$$\min_{\{\sigma_j\}_{j=1}^{J}} \ \frac{1}{N} \sum_{i=1}^{N} \left\| x^K(\xi^{(i)}, \{\sigma_j\}) - x^*(\xi^{(i)}) \right\|^2, \quad \xi^{(1)}, \dots, \xi^{(N)} \sim \mathcal{P}. \qquad \text{(ERM)}$$

Since $J$ is typically samll, the (ERM) is a small-scale unconstrained optimization problem, which can be solved efficiently by existing algorithms, such as SGD and ADAM. We note here that these commonly used optimizers rely on the back-propagation to compute the stochastic gradient estimators of the objective functions in (ERM). This implicitly requires that the computations in Algorithm 2 do not break the computational tree in order to keep track of the gradient information. Particularly, if the back-propagation does not fail when evaluating the proximal mapping of the function $g$, then (ERM) can be solved by stochastic gradient based optimizers. Otherwise, stochastic gradient based optimizers are no longer applicable. In this case, we may rely on grid search to find good penalty parameters, though it can be costly.

The main computational bottlenecks of the proposed L2O approach are summarized as follows: (1) Cost in obtaining the true solution $x^*$ which can be expensive and problem-depending. As a data-driven approach, we think this is an issue commonly seen in practice. (2) Cost in updating each block at each iteration of the MPALM. This can be relatively small since the MPALM is able to fully explore the multiblock structure of the underlying problem. (3) Cost in solving (ERM) in selecting the parameters $\{\sigma_j\}$. However, once a strategy is learned, it can be used for future tasks, which could save a significant amount of future computational cost.

For the rest of this section, we consider applying the proposed hyperparameter learning approach for solving two class of important problems that attract growing attention in recent years. This first problem is the Lasso problem, which plays an important role in compressed sensing and sparse coding. And the second problem is the discrete optimal transport problem, which has many applications in modern machine learning. Solving them efficiently has been and will remain an active research direction in the literature.

## 4.1 APPLICATION TO CLASSICAL LASSO PROBLEMS

Recall that the classical Lasso problem (Tibshirani, 1996) takes the following form:

$$\min_{w\in\mathbb{R}^n}\quad \frac{1}{2}\left\|Dw-\xi\right\|^2 + \mu\left\|w\right\|_1, \qquad\qquad (\text{Lasso}(\xi))$$

where $D\in\mathbb{R}^{m\times n}$ is the given dictionary, $\xi\in\mathbb{R}^m$ is a random variable sample from a fixed distribution $\mathcal{P}$, and $\mu>0$ denotes the regularization parameter. Though problem $(\text{Lasso}(\xi))$ is a special case of $(P(\xi))$ with $f_\xi(w):=\frac{1}{2}\left\|Dw-\xi\right\|^2$ and $g(w):=\mu\left\|w\right\|_1$, it is not desirable to apply Algorithm 1 for solving problem $(\text{Lasso}(\xi))$ directly. The reason is explained as follows. First, since $f_\xi$ is already a quadratic function, we can set $\Sigma = D^T D$ and get

$$q_\xi(w;w'):=f(w')+\langle\nabla f_\xi(w'),w-w'\rangle+\frac{1}{2}\left\|w-w'\right\|^2_{D^TD}=f_\xi(w),\quad\forall(w,w')\in\mathbb{R}^n\times\mathbb{R}^n.$$

Then, the objective function for the ALM subproblem in Line 2 of Algorithm 1 can be written as

$$\phi_{\xi,k}(w):=\frac{1}{2}\left\|Dw-\xi\right\|^2+\mu\left\|w\right\|_1+\frac{1}{2}\left\|w-w^k\right\|^2_\mathcal{S},$$

for a given $\mathcal{S}\in\mathbb{S}^n$. To ensure the convergence, the linear mapping $\mathcal{S}$ must satisfy that $\mathcal{S}+\frac{1}{2}D^TD\succ 0$. If one chooses $\mathcal{S}=\alpha I_n$ where $\alpha>0$ and $I_n$ denotes the identity matrix of size $n$, we see that the resulted algorithm is the same as the proximal point algorithm, and the ALM subproblem can still be challenging to solve since one needs to rely on a certain iterative scheme (see e.g., (Li et al., 2018a) for a more comprehensive study of this approach). On the other hand, if one chooses $\mathcal{S}=\alpha I_n-\frac{1}{2}D^TD$ with $\alpha>\frac{1}{2}\lambda_{\max}(D^TD)$, then solving the ALM subproblem is reduced to computing the proximal mapping of the function $g(w)=\mu\left\|w\right\|_1$, which admits analytical expression. In this case the algorithm coincides with the famous ISTA (Beck and Teboulle, 2009) which has been extensively studied in the literature. However, this approach requires evaluating $\lambda_{\max}(D^TD)$, which could be costly. Moreover, if $\alpha$ is chosen to be large, the convergence of the resulted algorithm can be quite slow, based on the empirical experience on the practical performance of the ISTA.

Motivated by the above arguments, we propose to solve $(\text{Lasso}(\xi))$ via solving its dual problem.

**Lemma 1 (Dual problem of ($\text{Lasso}(\xi)$))** *The dual problem of ($\text{Lasso}(\xi)$) is equivalent to the following minimization problem:*

$$\min_{y_1\in\mathbb{R}^n,\,y_2\in\mathbb{R}^m}\quad \delta_{\mathbb{B}_\mu}(-y_1)+\frac{1}{2}\left\|y_2\right\|^2-\langle\xi,y_2\rangle\quad\text{s.t.}\quad y_1+D^Ty_2=0. \qquad (\text{DLasso}(\xi))$$

*where $\mathbb{B}_\mu:=\{x\in\mathbb{R}^n\ :\ \left\|x\right\|_\infty\leq\mu\}$ denotes the $\infty$-norm ball in $\mathbb{R}^n$ with radius $\mu>0$.*

It is readily checked that the problem $(\text{DLasso}(\xi))$ is a special case of the general model $(P(\xi))$, and the objective function for the ALM subproblem can be written as

$$\phi_{\xi,k}(y):=\delta_{\mathbb{B}_\mu}(-y_1)+\frac{1}{2}\left\langle y,\begin{pmatrix}0&0\\0&I_m\end{pmatrix}y\right\rangle+\left\langle\begin{pmatrix}x^k\\Dx^k-\xi\end{pmatrix},y\right\rangle+\frac{\sigma}{2}\left\|\begin{pmatrix}I&D^T\end{pmatrix}y\right\|^2+\frac{1}{2}\left\|y-y^k\right\|^2_\mathcal{S},$$

where $\mathcal{S}$ is the SGS-operator of the associated $\mathcal{Q}$. Then, we see that $y^{k+1}=\operatorname{argmin}\{\phi_{\xi,k}(y)\ :\ y\in\mathbb{R}^n\times\mathbb{R}^m\}$ can be computed exactly as

$$y_2^{k+1/2}=\operatorname{argmin}\left\{\phi_{\xi,k}(y_1^k,y_2)\ :\ y_2\in\mathbb{R}^m\right\},$$

$$y_1^{k+1}=\operatorname{argmin}\left\{\phi_{\xi,k}(y_1,y_2^{k+1/2})\ :\ y_1\in\mathbb{R}^n\right\},$$

$$y_2^{k+1}=\operatorname{argmin}\left\{\phi_{\xi,k}(y_1^{k+1},y_2)\ :\ y_2\in\mathbb{R}^m\right\}.$$

Simple calculation shows that the update of $y_2$ involves solving a linear system with coefficient matrix $(I_m + \sigma DD^T)$ and the update of $y_1$ requires computing the proximal mapping of $g$ which is the projection operator onto the ball $\mathbb{B}_\mu$. Specifically, Algorithm 2 with $\mathcal{S}$ chosen to be the SGS-operator for $\mathcal{Q}$ given in the above applied for solving the problem (DLasso($\xi$)) can be summarized as in Algorithm 5. Readers are referred to Appendix D for the detailed description of the algorithm together with an efficient way of updating $\left(I_m + \sigma DD^T\right)^{-1}$.

Fixing the dictionary $D$, the initial point $(x^0, y_1^0, y_2^0)$, the step size $\tau$, and the maximum number of iterations, we can see that the output $x^K$ depends on the received signal $\xi$ and the penalty parameters $\{\sigma_j\}$. To emphasize the aforementioned dependency, we denote $x^K(\{\sigma_j\};\xi) := x^K$ as the output of the algorithm. Then, the learning objective is to solve the following ERM:

$$\min_{\{\sigma_j\}_{j=1}^J} \frac{1}{N}\sum_{i=1}^N \left\| x^K(\{\sigma_j\}, \xi^{(i)}) - x^*(\xi^{(i)}) \right\|^2, \quad \xi^{(i)} \sim \mathcal{P},\ i = 1,\dots,N. \tag{3}$$

### 4.2 APPLICATION TO OPTIMAL TRANSPORT PROBLEMS

Let $\xi := (\alpha;\beta) \in \mathbb{R}^m \times \mathbb{R}^n$ be a given tuple of two discrete probability distributions. Let $c \in \mathbb{R}^{m \times n}$ be a cost matrix. Then, the discrete optimal transport problem (Peyré et al., 2019) is stated as follows:

$$\min_{x \in \mathbb{R}^{m \times n}} \quad \langle c, x \rangle \quad \text{s.t.} \quad xe_n = \alpha,\ x^T e_m = \beta,\ x \geq 0. \tag{MOT($\xi$)}$$

Note the (4.2) is an instance of linear programming. By the duality theory for linear programming, the associated dual problem (as an equivalent minimization problem) is given as

$$\min_{y_1 \in \mathbb{R}^{m \times n},\ y_2 \in \mathbb{R}^m,\ y_3 \in \mathbb{R}^n} \quad \delta_+(y_1) - \langle \alpha, y_2 \rangle - \langle \beta, y_3 \rangle \quad \text{s.t.} \quad y_1 + y_2 e_n^T + e_m y_3^T = c, \tag{DOT($\xi$)}$$

which leads to the following objective function of the ALM subproblem by choosing appropriate $\mathcal{S}$:

$$\phi_{\xi,k}(y) := \delta_+(y_1) + \left\langle \begin{pmatrix} x^k \\ x^k e_n - \alpha \\ (x^k)^T e_m - \beta \end{pmatrix}, y \right\rangle + \frac{\sigma}{2} \left\| (I_{mn}\ e_n \otimes I_m\ I_n \otimes e_m) y - c \right\|^2 + \frac{1}{2} \left\| y - y^k \right\|_{\mathcal{S}}^2.$$

Consequently, we see from Theorem 2 that $y^{k+1} = \operatorname{argmin}\{\phi_{\xi,k}(y)\ :\ y \in \mathbb{R}^{m \times n} \times \mathbb{R}^n \times \mathbb{R}^m\}$ can be computed as (see also Algorithm 6 in Appendix E for the detailed description)

$$y_3^{k+1/2} = \operatorname{argmin}\left\{\phi_{\xi,k}(y_1^k, y_2^k, y_3)\ :\ y_3 \in \mathbb{R}^n\right\},$$

$$y_2^{k+1/2} = \operatorname{argmin}\left\{\phi_{\xi,k}(y_1^k, y_2, y_3^{k+1/2})\ :\ y_2 \in \mathbb{R}^m\right\},$$

$$y_1^{k+1} = \operatorname{argmin}\left\{\phi_{\xi,k}(y_1, y_2^{k+1/2}, y_3^{k+1/2})\ :\ y_1 \in \mathbb{R}^{m \times n}\right\},$$

$$y_2^{k+1} = \operatorname{argmin}\left\{\phi_{\xi,k}(y_1^{k+1}, y_2, y_3^{k+1/2})\ :\ y_2 \in \mathbb{R}^m\right\},$$

$$y_3^{k+1} = \operatorname{argmin}\left\{\phi_{\xi,k}(y_1^{k+1}, y_2^k, y_3)\ :\ y_3 \in \mathbb{R}^n\right\}.$$

Fixing the cost matrix $c \in \mathbb{R}^{m \times n}$, the initial point $(x^0, y_1^0, y_2^0, y_3^0)$, the step-size $\tau$ and the maximum number of iterations, we see that the output $x^K$ depends on the marginal distributions $\xi := (\alpha, \beta)$ and the penalty parameters $\{\sigma_j\}$. As usual, we denote $x^K(\{\sigma_j\}, \xi) := x^K$ as the output of the Algorithm 6. Then, the learning objective is to solve the following ERM:

$$\min_{\{\sigma_j\}_{j=1}^J} \frac{1}{N}\sum_{i=1}^N \left\| x^K(\{\sigma_j\}, \xi^{(i)}) - x^*(\xi^{(i)}) \right\|^2, \quad \xi^{(i)} := (\alpha^{(i)};\beta^{(i)}) \sim \mathcal{P},\ i = 1,\dots,N. \tag{4}$$

Obviously, the backpropagation with respect to $\{\sigma_j\}$ can be done in a straightforward manner.

## 5 RESULTS

In this section, we validate the practical efficiency of the learned MPALM (LMPALM) by applying it to solve Lasso problems and optimal transport problems. To facilitate comparison, we define

log-normalized mean squared error (NMSE) $\frac{1}{M}\sum_{i=1}^{M} 10\log_{10}\left(\frac{\|x_i - x_i^*\|_2}{\|x_i^*\|_2}\right)$, where $\{x_i^*\}_{i=1}^{M}$ and $\{x_i\}_{i=1}^{M}$ are the optimal solutions and the predicted solutions output by a certain algorithm associated with the testing data set, respectively. See Appendix F for detailed experimental settings.

For Lasso problems, We compare the numerical performance of the LMPALM with the MPALM algorithms using pre-specified penalty parameters and with the LISTA algorithm (Gregor and LeCun, 2010). The computational results with different choices of $(m, n)$ that plot the NMSE with respect to iteration numbers are presented in Figure 1. As anticipated, LMPALM outperforms fixed-parameter MPALM. Notably, we also observe that all MPALM-based algorithms admit linear convergence. In particular, LMPALM consistently shows faster convergence rate than the fixed-parameter alternatives. We expect this trend to continue with additional iterations, enhancing the comparative advantage of LMPALM. In comparison to LISTA, LMPALM also performs better. Indeed, LISTA exhibits very slow convergence rate in its early iterations, followed by rapid speed-up in later iterations. Conversely, LMPALM has a stable convergence rate, offering a clear advantage over LISTA's less uniform convergence pattern. Finally, we observe that when the problem size becomes larger, the problem becomes more difficult to solve and the accuracy of the returned solutions downgrades. In this case, $K = 64$ is not sufficient and more iterations are needed to get solutions with high accuracy.

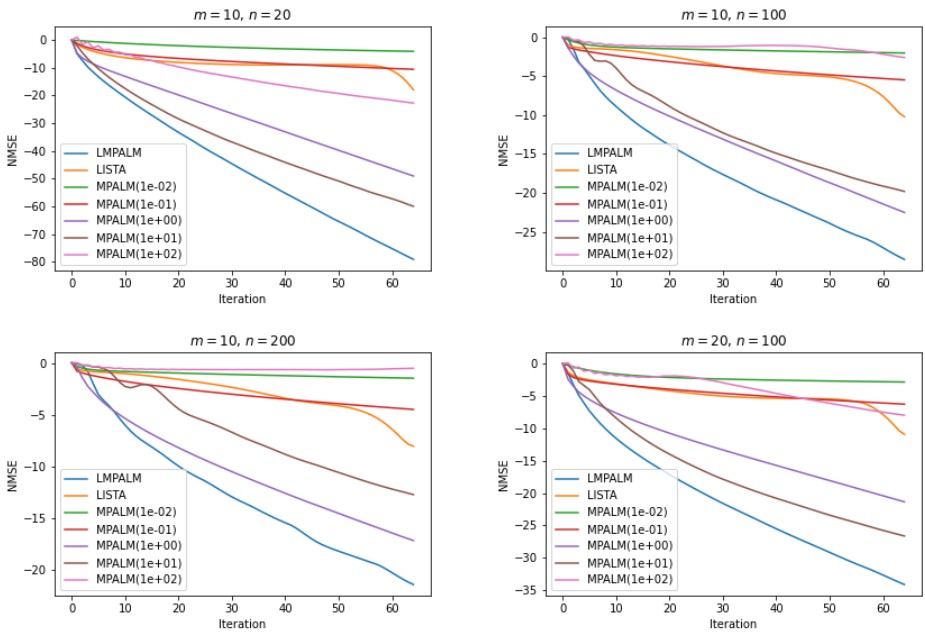

Figure 1: Lasso: normalized MSE for problem sizes $(m, n) = (10, 20)$, $(10, 100)$, $(10, 200)$, $(20, 100)$

For optimal transport problems, we compare the performance of LMPALM with the fixed-parameter MPALM algorithms and the commonly used Sinkhorn's algorithm for calculating approximate solutions (Cuturi, 2013z). Figure 2 displays the log-normalized mean-squared errors. From the results, we observe that the LMPALM algorithm empirically outperforms all fixed-parameter MPALM alternative and achieves a faster linear rate of convergence. When it comes to Sinkhorn's algorithm, we see that the accuracy of the solution is indeed highly sensitive to the entropy regularization parameter $\lambda$. Note that with a sufficiently small entropy parameter $\lambda$, Sinkhorn's method can approximate the solution to the optimal transport problem sufficiently well. However, small $\lambda$ results in slow convergence and can lead to some numerical issues. This limits the use of Sinkhorn's to find a highly accurate solution. Indeed, none of the four values of $\lambda$ offers a highly accurate solution to the optimal transport problem. On the contrary, the LMPALM approach shows excellent robustness. Lastly, to get higher accuracy, a larger number of iterations is needed.

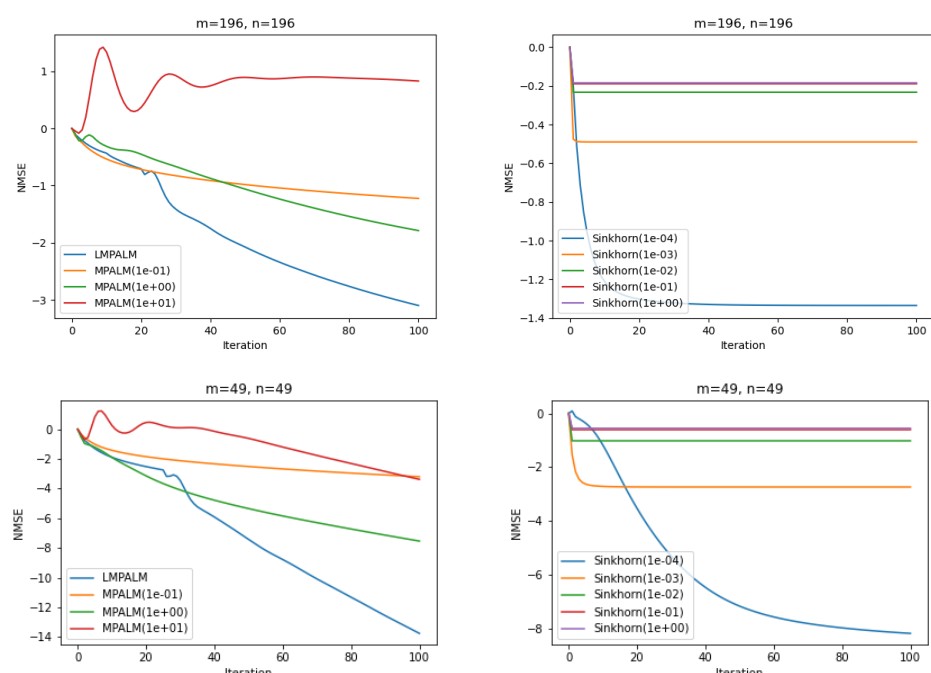

Figure 2: Optimal transport: NMSE for randomly generated data, with $m = n = 196$ and MNIST image data set, with $m = n = 49$. The first and third figures: LMPALM; The second and fourth figures: Sinkhorn's algorithm.

Our numerical findings suggest that data-driven approaches work well in practice because they have the potential to leverage the ability to learn patterns and structures from data without explicitly programming those patterns, thus providing appealing generalization capabilities from data.

## 6 CONCLUSION

In this paper, we successfully applied a Learning to Optimize (L2O) approach to hyperparameter learning for the Majorized Proximal Augmented Lagrangian Method (MPALM), a convergent multi-block ADMM-type method. Our approach has leveraged the convergence properties of MPALM while mitigating its detrimental sensitivity to the penalty hyperparameter. The computational results have demonstrated that MPALM, with adaptively trained hyperparameters, achieves faster convergence compared to existing alternatives in both Lasso and optimal transport problems. Notably, our algorithm's flexibility allows it to handle optimization problems with arbitrarily many block structures, motivating potential applications to more complex problems such as multi-marginal optimal transport (see Appendix E). However, it is also critical to acknowledge some current limitations of our work. All examples in this study involved Augmented Lagrangian Method (ALM) subproblems that could be solved exactly via elementary linear algebra routines, a requirement for implementing "autograd" for backpropagation. Future research could explore advanced techniques applicable to scenarios where subproblems are solved inexactly, broadening the applicability of our approach to a wider range of practical problems.

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

# Appendices

## A  THE TWO-BLOCK ADMM FOR PROBLEM $(P(\xi))$

By introducing an auxiliary variable $z$, we can reformulate the problem $(P(\xi))$ as:

$$\min_{y \in \mathbb{Y}, z \in \mathbb{Y}_1} f_\xi(y) + g(z), \quad \text{s.t.} \quad \mathcal{A}^* y = c, \ y_1 - z = 0.$$

Given $\xi$ and a penalty parameter $\sigma > 0$, the augmented Lagrangian function associated with the above problem can be written as

$$\mathcal{L}_{\xi,\sigma}(y, z; x, w) := f_\xi(y) + g(z) + \langle x, \mathcal{A}^* y - c \rangle + \langle w, y_1 - z \rangle + \frac{\sigma}{2} \|\mathcal{A}^* y - c\|^2 + \frac{\sigma}{2} \|y_1 - z\|^2.$$

Then, the two-block ADMM method can be described in Algorithm 3.

---

**Algorithm 3:** The classic two-block ADMM.

---

**Input:** A fixed point $\xi \in \Xi$, an initial point $(x^0, y^0, z^0, w^0)$, the penalty parameter $\sigma > 0$, the
    step size $\tau \in (0, \sqrt{1 + 5}/2)$, and the maximum number of iterations $K > 0$.

1 **for** $k = 0, \ldots, K - 1$ **do**
2     $y^{k+1} = \text{argmin} \mathcal{L}_{\xi,\sigma}(y, z^k; x^k, w^k)$.
3     $z^{k+1} = \text{argmin} \mathcal{L}_{\xi,\sigma}(y^{k+1}, z; x^k, w^k)$.
4     $x^{k+1} = x^k + \tau\sigma \left( \mathcal{A}^* y^{k+1} - c \right)$.
5     $w^{k+1} = w^k + \tau\sigma \left( y_1^{k+1} - z^{k+1} \right)$.
6 **end**
**Output:** $(x^K, y^K, z^K, w^K)$.

---

We can see that for updating $y$, we need to solve an optimization over the whole space $\mathbb{Y}$, which ignore the exploration of the separable structure of the problem.

## B  DIRECT MULTI-BLOCK EXTENSION OF THE TWO-BLOCK ADMM

Recall that the augmented Lagrangian function associated with the problem $(P(\xi))$ is defined as

$$\mathcal{L}_\sigma(y; x) := f_\xi(y) + g(y_1) + \langle \mathcal{A}^* y - c, x \rangle + \frac{\sigma}{2} \|\mathcal{A}^* y - c\|^2, \quad \forall(y, x) \in \mathbb{Y} \times \mathbb{X},$$

where $x \in \mathbb{X}$ denotes the Lagrange multiplier with respect to the linear constraint $\mathcal{A}^* y = c$. Then the multi-block ADMM has the following template, as shown in Algorithm 4.

---

**Algorithm 4:** The mutli-block ADMM.

---

**Input:** A fixed point $\xi \in \Xi$, an initial point $(x^0, y^0)$, the penalty parameter $\sigma > 0$, the step size
    $\tau \in (0, \sqrt{1 + 5}/2)$, and the maximum number of iterations $K > 0$.

1 **for** $k = 0, \ldots, K - 1$ **do**
2     **for** $i = 1, \ldots, p$ **do**
3        $y_i^{k+1} = \text{argmin} \mathcal{L}_{\xi,\sigma}(y_1^{k+1}, \ldots, y_{i-1}^{k+1}, y_i, y_{i+1}^k, \ldots, y_p^k; x^k)$.
4     **end**
5     $x^{k+1} = x^k + \tau\sigma \left( \mathcal{A}^* y^{k+1} - c \right)$.
6 **end**
**Output:** $(x^K, y^K)$.

---

However, as mentioned in the introduction, the above direct extension is not convergent unless under more stringent conditions (Chen et al., 2016).

## C  CONVERGENCE OF MPALM AND THE CHOICE OF $\mathcal{S}$

We first present the following condition that ensures the convergence of the MPALM, which is related to the solvability of the interested optimization problem.

**Assumption 2** *The first-order optimality conditions (1) admit at least one solution $(x^*, y^*) \in \mathbb{X} \times \mathbb{Y}$. Such a tuple is called a KKT solution.*

Assumption 2 is commonly employed in the literature, as it presented one of the weakest conditions ensuring the solvability of the problem $(P(\xi))$. Then, the global convergence of MPALM is presented as follows.

**Theorem 1 ((Chen et al., 2021))** *Suppose that Assumptions 1 and 2 hold and $\mathcal{S} : \mathbb{Y} \to \mathbb{Y}$ is a given self-adjoint linear operator such that $\frac{1}{2}\Sigma + \sigma\mathcal{A}\mathcal{A}^* + \mathcal{S} \succ 0$, and $\mathcal{S} \succeq -\frac{1}{2}\Sigma$. Let $\{(x^k, y^k)\}$ be the sequence generated by Algorithm 1. Then, the sequence is bounded and converges to a KKT solution.*

Next, we shall provide more details on how to choose the self-adjoint operator in order to define the proximal term in MPALM. To this end, we consider the mapping $\mathcal{S}$ which can be decomposed as the sum of two self-adjoint mappings, i.e., $\mathcal{S} := \widetilde{\mathcal{S}} + \widehat{\mathcal{S}}$ where $\widetilde{\mathcal{S}} := \mathrm{Diag}(\widetilde{\mathcal{S}}_{11}, \ldots, \widetilde{\mathcal{S}}_{pp})$ is block-diagonal with $\widetilde{\mathcal{S}}_{ii} : \mathbb{Y}_i \to \mathbb{Y}_i$, for $i = 1, \ldots, p$, and $\widehat{\mathcal{S}}$ to be determined shortly. Recall that the objective function for the ALM subproblem in Algorithm 1 can be simplified as

$$\phi_{\xi,k}(y) = \frac{1}{2}\left\langle y, (\sigma\mathcal{A}\mathcal{A}^* + \mathcal{S} + \Sigma)\, y \right\rangle + \left\langle \nabla f_\xi(y^k) + \mathcal{A}x^k - \sigma\mathcal{A}c - \mathcal{S}y^k - \Sigma y^k, y \right\rangle + g(y_1) + \text{const}, \tag{5}$$

where "const" means a quantity that does not depend on $y$. For later usage, we write $\mathcal{Q} := \sigma\mathcal{A}\mathcal{A}^* + \widetilde{\mathcal{S}} + \Sigma := \mathcal{U} + \mathcal{D} + \mathcal{U}^*$, where $\mathcal{D} : \mathbb{Y} \to \mathbb{Y}$ and $\mathcal{U} : \mathbb{Y} \to \mathbb{Y}$ are defined as

$$\mathcal{D} := \mathrm{Diag}\left(\sigma\mathcal{A}_1\mathcal{A}_1^* + \Sigma_{11} + \widetilde{\mathcal{S}}_{11}, \ldots, \sigma\mathcal{A}_p\mathcal{A}_p^* + \Sigma_{pp} + \widetilde{\mathcal{S}}_{pp}\right),$$

$$\mathcal{U} := \begin{pmatrix} 0 & \sigma\mathcal{A}_1\mathcal{A}_2^* + \Sigma_{12} & \ldots & \sigma\mathcal{A}_1\mathcal{A}_{p-1}^* + \Sigma_{1,p-1} & \sigma\mathcal{A}_1\mathcal{A}_p^* + \Sigma_{1p} \\ 0 & 0 & \ldots & \sigma\mathcal{A}_2\mathcal{A}_{p-1}^* + \Sigma_{2,p-1} & \sigma\mathcal{A}_2\mathcal{A}_p^* + \Sigma_{2p} \\ \vdots & \vdots & \ddots & \vdots & \vdots \\ 0 & 0 & \ldots & 0 & \sigma\mathcal{A}_{p-1}\mathcal{A}_p^* + \Sigma_{p-1,p} \\ 0 & 0 & \ldots & 0 & 0 \end{pmatrix}.$$

For the choice of $\widetilde{\mathcal{S}}$, we only require that the following assumption holds.

**Assumption 3 (Positive definiteness of $\mathcal{Q}$)** $\widetilde{\mathcal{S}} = \mathrm{Diag}(\widetilde{\mathcal{S}}_{11}, \ldots, \widetilde{\mathcal{S}}_{pp})$ *is chosen appropriately such that*

$$\frac{1}{2}\Sigma_{ii} + \sigma\mathcal{A}_i\mathcal{A}_i^* + \widetilde{\mathcal{S}}_{ii} \succ 0, \ i = 1, \ldots, p, \quad \widetilde{\mathcal{S}} \succeq -\frac{1}{2}\Sigma$$

Under Assumption 3, $\mathcal{D}$ is positive definite and hence nonsingular. Using the above decomposition, we propose to choose $\widehat{\mathcal{S}} := \mathcal{U}\mathcal{D}^{-1}\mathcal{U}^*$, which is called the SGS-operator for $\mathcal{Q}$, denoted by $\mathrm{SGS}(\mathcal{Q})$. With this particular choice of $\widehat{\mathcal{S}}$, we can show in the following theorem that one cycle of the block symmetric Gauss-Seidel update exactly solves the ALM subproblem in Line 2 of Algorithm 1.

**Theorem 2 (Minimizing the ALM subproblem (Li et al., 2019, Theorem 1))** *Under Assumption 3, the minimizer*

$$y^{k+1} = \mathrm{argmin}\{\phi_{\xi,k}(y) \ : \ y \in \mathbb{Y}\}$$

*can be computed exactly as follows:*

$$\tilde{y}_i^k = \mathrm{argmin}\left\{ \mathcal{L}_{\xi,\sigma}(y_{<i}^k, y_i, \tilde{y}_{>i}^k; x^k, y^k) + \frac{1}{2}\left\| y_i - y_i^k \right\|_{\widehat{\mathcal{S}}_{ii}}^2 \ : \ y_i \in \mathbb{Y}_i \right\}, \quad i = p, \ldots, 2,$$

$$y_i^{k+1} = \mathrm{argmin}\left\{ \mathcal{L}_{\xi,\sigma}(y_{<i}^{k+1}, y_i, \tilde{y}_{>i}^k; x^k, y^k) + \frac{1}{2}\left\| y_i - y_i^k \right\|_{\widehat{\mathcal{S}}_{ii}}^2 \ : \ y_i \in \mathbb{Y}_i \right\}, \quad i = 1, \ldots, p.$$

---

**Algorithm 5:** The proximal ALM for (DLasso($\xi$))

---

**Input:** The dictionary $D \in \mathbb{R}^{m \times n}$, the received signal $\xi \in \mathbb{R}^m$, the regularization parameter
$\mu > 0$, an initial point $(x^0, y_1^0, y_2^0) \in \mathbb{R}^n \times \mathbb{R}^n \times \mathbb{R}^m$, the maximum number of iterations
$K > 0$, a positive integer $K_0 \leq K$, and the set of penalty parameters
$\{\sigma_j \ : \ 0 \leq j \leq \lfloor K/K_0 \rfloor + 1\}$.

1 **for** $k = 0, \ldots, K - 1$ **do**
2     Find $j$ such that $k \in [jK_0, (j+1)K_0)$ and set $\sigma = \sigma_j$.
3     $y_2^{k+1/2} = \left(I_m + \sigma DD^T\right)^{-1} \left(\xi - Dx^k - \sigma Dy_1^k\right)$.
4     $y_1^{k+1} = -\text{proj}_{\mathbb{B}_\mu}\left(D^T y_2^{k+1/2} + \frac{1}{\sigma}x^k\right)$.
5     $y_2^{k+1} = \left(I_m + \sigma DD^T\right)^{-1}\left(\xi - Dx^k - \sigma Dy_1^{k+1}\right)$.
6     $x^{k+1} = x^k + \tau\sigma\left(y_1^{k+1} + D^T y_2^{k+1}\right)$.
7 **end**

**Output:** $x^K$.

---

## D  MPALM FOR LASSO

The MPALM applied for solving the dual form of the Lasso problem is presented in Algorithm 5.

We next show how to efficiently update the inverse of the matrix $\left(I_m + \sigma DD^T\right)$ without breaking down the computational tree when performing backpropagation as needed in optimizing (3). Let the spectral decomposition of $DD^T$ be given as

$$DD^T = P\Lambda P^T, \quad \Lambda = \text{Diag}(\lambda_1, \ldots, \lambda_m), \quad \lambda_1 \geq \cdots \geq \lambda_m \geq 0.$$

Then for any $\sigma > 0$, we see that the matrix $I_m + \sigma DD^T$ admits a spectral decomposition

$$I_m + \sigma DD^T = P\text{Diag}\left(1 + \sigma\lambda_1, \ldots, 1 + \sigma\lambda_m\right)P^T.$$

Hence, we get

$$\left(I_m + \sigma DD^T\right)^{-1} = P\text{Diag}\left(\frac{1}{1 + \sigma\lambda_1}, \ldots, \frac{1}{1 + \sigma\lambda_m}\right)P^T.$$

Here, the orthogonal matrix $P$ and the eigenvalues $\lambda_1, \ldots, \lambda_m$ need only to be computed once. So, the inverse $\left(I_m + \sigma DD^T\right)^{-1}$, as a function of $\sigma$, is continuously differentiable in $\sigma$.

## E  MPALM FOR OPTIMAL TRANSPORT

The MPALM applied for solving the dual problem of optimal transport problem can be described in Algorithm 6.

**Remark 1** *The proposed framework can be easily extended to solve the multi-marginal optimal transport problems (Mehta et al., 2023):*

$$\min_{x \in \mathbb{R}^{n_1 \times \cdots \times n_q}} \langle c, x \rangle \quad \text{s.t.} \quad \mathcal{A}_i(x) = \alpha^{(i)}, \quad i = 1, \ldots, q, \ x \geq 0,$$

*where $c \in \mathbb{R}^{n_1 \times \cdots \times n_q}$ is the cost tensor, $\mathcal{A}_i : \mathbb{R}^{n_1 \times \cdots \times n_q} \to \mathbb{R}^{n_i}$ is a linear mapping that compute the $i$-th marginal of its input, and $\alpha^{(i)} \in \mathbb{R}^{n_i}$ are given marginal distribution, for $i = 1, \ldots, q$. The corresponding dual problem (as an equivalent minimization problem) is then given by*

$$\min_{y_i \in \mathbb{R}^{n_i}, \ 1 \leq i \leq q} \delta_+(y_1) - \sum_{i=1}^{q}\left\langle \alpha^{(i)}, y_{i+1}\right\rangle \quad \text{s.t.} \quad y_1 + y_2 \oplus \cdots \oplus y_{q+1} = c,$$

*where $y_2 \oplus \cdots \oplus y_{q+1} \in \mathbb{R}^{n_1 \times \cdots \times n_q}$ denotes the tensor whose $(i_2, \ldots, i_{q+1})$ entry is $y_2(i_2) + \cdots + y_{q+1}(i_{q+1})$, for any indices $1 \leq i_2 \leq n_1, \ \ldots, 1 \leq i_{q+1} \leq n_q$. We see that the dual problem has $q + 1$ blocks and the proposed methodology is directly applicable.*

---

**Algorithm 6:** The proximal ALM for (DOT($\xi$))

---

**Input:** Two marginal distributions $\xi := (\alpha; \beta) \in \mathbb{R}^m \times \mathbb{R}^n$, the cost matrix $c \in \mathbb{R}^{m \times n}$, an initial point $(x^0, y_1^0, y_2^0, y_3^0) \in \mathbb{R}^{m \times n} \times \mathbb{R}^{m \times n} \times \mathbb{R}^m \times \mathbb{R}^n$, the maximum number of iterations $K > 0$, a positive integer $K_0 \leq K$, and the set of penalty parameters $\{\sigma_j \ : \ 0 \leq j \leq \lfloor K/K_0 \rfloor + 1\}$.

**1 for** $k = 0, \dots, K - 1$ **do**

2     Find $j$ such that $k \in [jK_0, (j+1)K_0)$ and set $\sigma = \sigma_j$.

3     $y_3^{k+1/2} = \frac{1}{m}\left(\frac{1}{\sigma}\beta - \left(y_1^k + y_2^k e_n^T - c + \frac{1}{\sigma}x^k\right)^T e_m\right).$

4     $y_2^{k+1/2} = \frac{1}{n}\left(\frac{1}{\sigma}\alpha - \left(y_1^k + e_m(y_3^{k+1/2})^T - c + \frac{1}{\sigma}x^k\right)e_n\right).$

5     $y_1^{k+1} = \max\left\{0, c - y_2^{k+1/2}e_n^T - e_m(y_3^{k+1/2})^T - \frac{1}{\sigma}x^k\right\}.$

6     $y_2^{k+1} = \frac{1}{n}\left(\frac{1}{\sigma}\alpha - \left(y_1^{k+1} + e_m(y_3^{k+1/2})^T - c + \frac{1}{\sigma}x^k\right)e_n\right).$

7     $y_3^{k+1} = \frac{1}{m}\left(\frac{1}{\sigma}\beta - \left(y_1^{k+1} + y_2^{k+1}e_n^T - c + \frac{1}{\sigma}x^k\right)^T e_m\right).$

8     $x^{k+1} = x^k + \tau\sigma\left(y_1^{k+1} + y_2^{k+1}e_n^T + e_m(y_3^{k+1})^T - c\right).$

**9 end**

**Output:** $x^K$.

---

## F    Detailed experimental settings

We shall present the detailed experimental setting for both applications.

### F.1   Lasso problems

It is known that L2O has demonstrated promising performance in the Lasso problem. One notable L2O approach, extensively studied in the literature, is the Learned ISTA (LISTA) (Gregor and LeCun, 2010). LISTA, similar in structure to ISTA but with trained parameters, exhibits substantial improvements over its predecessor. Recent advancements have been made to further improve the convergence properties of the approach, however, the improvement in terms of practical efficiency is relatively modest, based on the results presented in (Chen et al., 2022) and references therein. We focus on demonstrating that our proposed algorithm significantly outperforms the original LISTA, thereby outperforming similar approaches that share comparable performance with LISTA.

Following a similar procedure in (Chen et al., 2022), we consider testing randomly generated data. Particularly, in our experiment, the regularization parameter $\mu$ is fixed to be $0.1$ and $D \in \mathbb{R}^{m \times n}$ is generated by sampling its entries from the standard Gaussian distribution and then normalizing the columns to have unit norms. With given $\mu$ and $D$, we then sample $50,000$ vectors $\{\xi^{(i)}\}_{i=1}^N$ from the standard Gaussian distribution to generate a set of Lasso problems. The optimal solutions $\{x_i^*\}_{i=1}^N$ for these problems are then computed using the powerful commercial solver GUROBI (version 11.0.1, with an academic license). The training size is set as $N = 45,000$ and the testing size is set as $M = 5,000$.

We compare the numerical performance of the LMPALM with the MPALM algorithms using pre-specified penalty parameters $\sigma = 10^k$, where $k = -2, -1, 0, 1, 2$, and with the LISTA algorithm (Gregor and LeCun, 2010). The computational results with different choices of $(m, n)$ that plot the NMSE with respect to iteration numbers are presented in Figure 1, where the maximum number of iterations is set as $K = 64$. Both LMPALM and LISTA are trained to minimize objective ERM using the AdamW optimizer. Hyperparameters for the training of LISTA are taken from (Liu and Chen, 2019). For LMPALM, we set betas $= (0.999, 0.999)$ and learning rate as lr $= 0.001$ [1]. Moreover, in LMPALM, we use eight restarts, i.e., a set of eight parameters $\{\sigma_j\}_{j=1}^8$ is learned. The parameters are initialized as 1. Based on our numerical experience, increasing the number of restarts enhances the robustness of the algorithm, albeit with increased computational cost. A batch size of $4,500$ is used in our training and the model is trained for 250 epochs.

---

[1]See https://pytorch.org/docs/stable/generated/torch.optim.AdamW.html for more detials.

## F.2 OPTIMAL TRANSPORT PROBLEMS

Next, we shall consider the optimal transport problem. In our experiments, we set $m = n$ for simplicity. The cost matrix $c \in \mathbb{R}^{m \times n}$ captures the squared distance between corresponding entries of $\alpha$ and $\beta$, i.e., $C_{ij} = |i - j|^2$ for $1 \leq i, j \leq m$.

We consider the following two ways for generating the data set $\{(\alpha^{(i)}, \beta^{(i)})\}_{i=1}^{5000}$ consisting of 5,000 instances: (1) the marginal distributions $\alpha^{(i)}$ and $\beta^{(i)}$ are randomly generated whose entries are drawn from the uniform distribution on the interval $(0, 1)$; (2) $\alpha^{(i)}$ and $\beta^{(i)}$ are generated by flattening $7 \times 7$ MNIST images corresponding to the digits "2" and "4", respectively. Note that we also normalized the distributions $\alpha^{(i)}$ and $\beta^{(i)}$ by dividing their sums so that they all sum up to one. Again, for each case, the corresponding true optimal solution is computed by the commercial solver GUBORI. The training size and the testing size are set as $N = 4,500$ and $M = 500$, respectively. We train LMPALM with four restarts for 500 epochs with a batch size of 2,750. As usual, the initial parameters $\{\sigma_j\}_{j=1}^{4}$ are initialized as one. For MPALM-based methods, we set the total number of iterations as $K = 100$.

