# OpenReview forum: "Accelerating Multi-Block Constrained Optimization Through Learning to Optimize"
_ICLR.cc/2025/Conference — ICLR 2025 Conference Withdrawn Submission_

### Official Review · Reviewer_3Kwo · 2024-10-29

**Soundness:** 3
**Presentation:** 2
**Contribution:** 2
**Rating:** 5
**Confidence:** 3

**Summary:**

The paper proposes an effective Learning to Optimize (L2O) framework for hyperparameter tuning in a majorized proximal augmented Lagrangian method, designed to solve challenging optimization problems of the form $P(\xi)$. This work expands the application of machine learning techniques in developing efficient algorithms for constrained optimization. The framework is validated through numerical experiments on the Lasso problem and the discrete optimal transport problem.

**Strengths:**

1.Writing: This work is presented with good writing style, where the summarized problems with detailed explanations make it easy for readers to understand the problem addressed in this article.

2.Novelty:It proposes a simple yet effective L2O framework for learning the hyperparameter of a majorized proximal augmented Lagrangian method.

3.Experiments: Several scenarios and recent baselines are considered, implying improvements in accuracy and robustness under various distributional shifts.

**Weaknesses:**

Weaknesses:
1.Originality of the algorithm. Could you clarify the differences between the algorithm presented in this paper and the one in [1]?  If the framework proposed here is indeed more inclusive than MPALM, please provide an example of a problem that this paper's algorithm can solve but MPALM cannot.

[1]Liang Chen, Xudong Li, Defeng Sun, and Kim-Chuan Toh. On the equivalence of inexact proximal ALM and ADMM for a class of convex composite programming. Mathematical Programming,185(1-2):111–161, 2021.

2.Lack of strong theoretical guarantees. There is no specific convergence analysis for the algorithm, nor is there a provided convergence rate. The main text includes two theorems, one of which cites conclusions from other sources. If convergence analysis is challenging, a detailed explanatory analysis should be provided.

3.Writing. The paper seems to lack a detailed introduction to LMPALM, as it first appears in abbreviated form in the RESULTS section. Is this an oversight in the writing? Additionally, the RESULTS section does not clearly explain why LMPALM improves the experimental results.

4.Computational Cost. The paper may not provide a comprehensive assessment of the computational efficiency and practicality of the proposed method in real-world applications. Like the computational complexity analysis or empirical time/memory cost.

**Questions:**

For specific questions, please refer to the "Weakness" section.
I would be willing to raise my score if these questions concerned are well addressed.

---

> ### Author Response · Authors · 2024-11-24
> **Authors' response to the Reviewer 3Kwo**
>
> Thank you very much for the valuable comments and suggestions. Below is our response.
>
> ## Originality of the Algorithm
> We appreciate the reviewer's request for clarification on the originality of our algorithm. The framework presented in this paper **extends the MPALM framework proposed in [1] by incorporating a novel Learning to Optimize (L2O) approach to handle multi-block constrained optimization problems, a domain where traditional L2O methods face limitations**. Our framework is more inclusive in the sense that it **adapts parameters dynamically through learning**, which **enhances flexibility and performance MPALM**. This is indeed supported by our numerical experiments, which motivates further potential applications in various domains that can be modeled as constrained multi-block optimization problems.
>
> ## Lack of Strong Theoretical Guarantees and Detailed Computational Cost
> We acknowledge the reviewer’s concern regarding the lack of specific convergence guarantees. While the primary focus of this paper is to introduce a novel L2O framework for addressing multi-block constrained optimization problems, we agree that establishing rigorous theoretical guarantees for the **adaptive/restarted MPALM** is crucial for fully validating the approach. We regret that we are unable to provide comprehensive theoretical results at this stage. However, we view this work as an important **initial step** in this research direction, laying the groundwork for future studies to develop and formalize these guarantees.
>
> We appreciate the reviewer's request for a more thorough assessment of the computational cost and practicality of our method. However, we shall argue that since the proposed framework is quite general, and the computational cost is **highly problem dependent**. The **main computational bottlenecks** are: (1) Cost in obtaining the true solution $x*$ which can be expensive and problem-depending. As a data-driven approach, we think this is an issue commonly seen in practice. (2) Cost in updating each block at each iteration of the MPALM. This can be relatively small, since the MPALM is able to fully explore the multi-block structure of the underlying problem. (3) Cost in solving the optimization in selecting the parameters $\{\sigma_j\}$. However, once a strategy is learned, it can be used for future tasks, which could save a significant amount of future computational cost.  We have placed the above discussions in the revised paper.
>
> ## Presentation
> We apologize for the lack of detailed definitions to MPALM and LMPALM in the main text. This was an oversight, and we will revise the manuscript to address these issues. We also add further insights on why LMPALM performs betters at the end of the paper. **Data-driven approaches work well because they have the potential to leverage the ability to learn patterns and structures from data without explicitly programming those patterns, thus providing appealing generalization capabilities from data.**

---

### Official Review · Reviewer_oDt7 · 2024-10-31

**Soundness:** 2
**Presentation:** 2
**Contribution:** 2
**Rating:** 5
**Confidence:** 3

**Summary:**

The authors learn parameters in the optimisation algorithm MPALM, as the performance of the algorithm is shown to be highly sensitive to penalty parameter choice. The authors learn penalty parameters using unrolling with a regression loss. The authors apply their learned algorithm to Lasso regression and an optimal transport problem.

**Strengths:**

Originality of learned parameters for MPALM algorithm. Learned algorithm shows an increased performance over commonly used algorithms for both lasso and optimal transport problems, as well as MPALM with a variety of chosen penalty parameters. Good summary of related research and past work on L2O. Comprehensive introduction to the MPALM algorithm.

**Weaknesses:**

For the LASSO problem FISTA is a good option due to acceleration, however, it isn't compared. Not enough novelty, only parameters are learned, no novel algorithm, problem, or method of learning parameters. Theorem 1 is stated without reference or proof. While provable convergence is stated, it lacks a proof as to why, as the condition (Assumption 3) depends on \sigma.

Minor: Grammatical errors/typos including "with slightly abuse of notation", indexing with i in (ERM). "g(w) := \mu \| x \|_1", "the Sinkhorn’s algorithm". It seems MPALM and LMPALM are never explicitly defined in the main text.

**Questions:**

Is \{\sigma_j\} supposed to denote \{\sigma_1, \cdots, \sigma_{J}\}, where $J = \floor{K/K_0}+1$? \{\sigma_j\} seems ill-defined without indexing j. With Assumption 3, does convergence with learned parameters hold for your learned algorithms. Is this a clear and easy result? You write "If the back-propagation fails in practice, stochastic gradient based optimizers are no longer applicable. In this case, we may rely on grid search to find good penalty parameters, though it can be costly." Which cases is this true for?

---

> ### Author Response · Authors · 2024-11-24
> **Authors' response to the Reviewer oDt7**
>
> Thank you for your thorough review and valuable feedback. Below, we address the weaknesses, minor issues, and questions you have raised.
>
> ##  Comparison with FISTA
> We acknowledge that FISTA is a strong baseline for the LASSO problem due to its acceleration properties. However, based on our numerical experience and the numerical results provided in [Figure 6, Chen et al., 2022], one observes that **LISTA outperforms FISTA empirically**. For simplicity, we refrain from presenting the computational results for FISTA and focusing on comparing L2O approaches.
>
> ##  Novelty of the Work
> While the proposed framework focuses on learning parameters rather than introducing a completely new algorithm, we argue that **the novelty lies in developing a unified learn-to-optimize (L2O) ADMM-type framework tailored for multi-block constrained optimization problems**. These problems are **complex and underexplored** in the current literature. Furthermore, the L2O approach introduces a novel perspective on how **machine learning approaches can enhance the efficiency of existing optimization methods**.
>
> ## Convergence
> We apologize for the omission of references and proofs for Theorem 1. In the revised version, **Theorem 1 have been properly referenced**. We also understand the concern regarding Assumption 3 and its role in provable convergence. **The convergence for the adaptive MPALM remains an open question, which we defer in future research**.
>
> ## Typos
> We appreciate you pointing out the grammatical issues. These have been corrected in the revised manuscript to improve clarity and readability.
>
> ## Issues in Back-Propagation
> The statement regarding back-propagation failing in practice applies to scenarios where gradient-based methods struggle, such as cases with **non-differentiable operations**. We clarify that if the updating rule for each block only involves **almost surely differentiable operations**, such as ReLU, then our framework will be applicable through back-propagation in modern libraries such as PyTorch and JAX.
>
>
>
> Chen T, Chen X, Chen W, Heaton H, Liu J, Wang Z, Yin W. Learning to optimize: A primer and a benchmark. Journal of Machine Learning Research. 2022;23(189):1-59.

---

> > ### Comment · Reviewer_oDt7 · 2024-12-01
> >
> > Thank you for answering my comments. I will keep my score the same.

---

### Official Review · Reviewer_SKcj · 2024-11-02

**Soundness:** 1
**Presentation:** 1
**Contribution:** 2
**Rating:** 3
**Confidence:** 2

**Summary:**

This work addresses a gap in Learning to Optimize (L2O) approaches by extending L2O methods to multi-block ADMM-type algorithms, which remain unexplored. Multi-block methods take advantage of the separable structure of optimization problems, reducing per-iteration complexity. However, classical multi-block ADMM lacks convergence guarantees, making the Majorized Proximal Augmented Lagrangian Method (MPALM) a more suitable choice due to its convergence properties. Despite its theoretical benefits, MPALM's performance is highly sensitive to the choice of penalty parameters. To address this, this work proposes a novel L2O framework that adaptively learns these hyperparameters through supervised learning, demonstrating its effectiveness on the Lasso problem and the optimal transport problem.

**Strengths:**

1. This work proposes a simple yet effective L2O framework for learning the hyperparameters of a majorized proximal augmented Lagrangian method.

2. Two applications, i.e., Lasso problem and discrete optimal transport problem are used to evaluate the proposed framework.

**Weaknesses:**

I'm not an expert in this field, but I have provided some review comments based on my experience. I will consider the feedback from other reviewers and the authors' rebuttal to adjust my score. I have some concerns as follows.

1. The key contribution of this work lies in the proposed L2O framework for majorized proximal augmented Lagrangian method. However, this content only appears when reading page 7, as the authors dedicate extensive space to introducing majorized proximal augmented Lagrangian method. I have a few questions: 1) Are Theorem 1 and Theorem 2 intended as theoretical contributions in this work? If not, could you clarify why they are presented as main text theorems? Additionally, I could not locate proofs for these theorems within the manuscript. 2) While the asymptotic convergence for majorized proximal augmented Lagrangian method is discussed, I am particularly interested in insights regarding the non-asymptotic convergence for majorized proximal augmented Lagrangian method.

2. The applications demonstrated in this work are the Lasso problem and the discrete optimal transport problem. Could the proposed method also be applied to popular machine learning tasks?

3. One of the key contributions of this work is the introduction of a hyperparameter learning framework. Bilevel optimization is also widely used for hyperparameter optimization in machine learning. Could bilevel optimization methods be employed to optimize the hyperparameters in this work as well?

4. If Theorems 1 and 2 are not the theoretical contributions of this paper, then what are the theoretical contributions? I did not find any other theorems in the paper.

5. Could you modify Figures 1 and 2? The font size is too small and doesn’t look very appealing.

**Questions:**

Please refer to the Weaknesses.

---

> ### Author Response · Authors · 2024-11-23
> **Authors' response to the Reviewer SKcj**
>
> Thank you for your thoughtful review and feedback, which have helped to improve the quality of the paper. Below are our responses to your points:
>
> ## Presentation
> In the revised manuscript, we will streamline the introduction of MPALM and **move some background information to the appendix**. This will allow us to present the key contributions earlier in the paper, providing readers with a clearer understanding of the proposed method and its significance. We would also like to thank you for pointing out the issue with the font size and visual appeal of Figures 1 and 2. We will **redesign these figures with larger, more legible fonts and an improved layout to enhance clarity and visual quality**.
>
> ## Theoretical Contribution of Theorems 1 and 2
> Theorems 1 and 2 are foundational results that establish the convergence properties of MPALM and how we can fully explore the block structure of the optimization problem, a core building block of our proposed L2O framework. **These results are crucial to the overall framework, but are not novel in themselves**. We included them in the main text to provide a complete and coherent description of MPALM. However, we will **revise the manuscript to clearly distinguish these results from our original contributions and explicitly reference their sources**.
>
> ##	Non-Asymptotic Convergence
> Under a suitable error-bound condition, one can establish the **linear convergence rate of the method in terms of Karush-Kuhn-Tucker (KKT) residues**; see [Chen et al., 2021]. However, we shall mention that for many practical problems, the error-bound condition cannot be verified before the optimal solution has been computed.
>
> ##	Theoretical Contributions Beyond Theorems 1 and 2
> The primary contribution of this paper is **the introduction of the L2O framework tailored for multi-block constrained optimization problems, specifically using MPALM as the foundation**. While Theorems 1 and 2 describe properties of MPALM, our original contribution lies in formulating a learn-to-optimize framework that can generalize across problem instances. One interesting future direction will be **analyzing the convergences restarted/adaptive variants of the MPALM**. For instance, one may try to characterize conditions on ${\sigma_j\}$ so that the restarted/adaptive is convergence.
>
> ## Applicability to Popular Machine Learning Tasks
> The proposed method has the potential to be extended to popular machine learning tasks. Although our experiments focus on the Lasso and discrete optimal transport problems to validate the framework, its flexibility allows for broader applications **as long as one is able to formulate the problem as the constrained multi-block convex optimization problems**, including the multi-marginal optimal transport problems that are challenging to solve by existing methods in the literature.
>
> ## Use of Bilevel Optimization for Hyperparameter Learning
> We appreciate the suggestion to explore bilevel optimization for hyperparameter learning. While the current work does not mention the bilevel optimization explicitly, we acknowledge its potential relevance. We refrain from adding related discussions on the bilevel optimization approach since we want to **focus more on L20 for ADMM-type algorithms and its applications**.
>
> Liang Chen, Xudong Li, Defeng Sun, and Kim-Chuan Toh. On the equivalence of inexact proximal ALM and ADMM for a class of convex composite programming. Mathematical Programming,185(1-2):111–161, 2021.

---

> > ### Comment · Reviewer_SKcj · 2024-11-25
> >
> > Thank you for the responses. However, my concerns have not been fully addressed, e.g., 1) I still have some concerns about the theoretical contributions of this work; 2) the applicability of this work to popular machine learning tasks is still unclear; 3) The readability of the manuscript needs further improvement.  After considering the rebuttals and the comments from all reviewers, I have decided to update my score to 3.

---

> ### Author Response · Authors · 2024-11-25
>
> Thank you for your prompt reply.
>
> Regarding the theoretical guarantees, we appreciate your comments and have addressed them in the response to the reviewer YmSS. For the applicability of this work, we would like to highlight an example in federated learning, which involves solving optimization problems of the form:
> $$
> \min_{x} \frac{1}{N}\sum_{i=1}^N f_i(x) + r(x)
> $$
> which can be formulated as
> $$
> \min_{x, x_1,\dots, x_N} \frac{1}{N}\sum_{i=1}^N f_i(x_i) + r(x) \quad \mathrm{s.t.} \quad x = x_i,  i = 1,\dots N.
> $$
> This reformulation aligns with the multiblock constrained optimization framework studied in our work. For further details and applications of federated learning, we refer you to the review article:
>
> Li L, Fan Y, Tse M, Lin KY. A review of applications in federated learning. Computers & Industrial Engineering. 2020 Nov 1;149:106854.
>
> Regarding the readability of the paper, we would greatly appreciate your further feedback on specific areas that could be improved. While we have endeavored to address all the issues raised by the reviewers, we welcome any additional suggestions to enhance the clarity and presentation of our work.
>
> Thank you for your constructive input, and we look forward to hearing your thoughts.

---

### Official Review · Reviewer_YmSS · 2024-11-03

**Soundness:** 2
**Presentation:** 2
**Contribution:** 1
**Rating:** 3
**Confidence:** 3

**Summary:**

This paper studies a novel learning to optimize approach for multi-block ADMM. The proposed method adaptively selects the penaltiy parameters for MPALM and achieve better empirical performance.

**Strengths:**

The studied problem is interesting. The paper is easy to follow. The proposed method obtains better peroformance than baselines in the Lasso problem and the discrete optimal transport problem.

**Weaknesses:**

Here are my concerns about this paper:

- About the (ERM)

My main concern is that it is usually expensive or intractable to obtain the exact x* for (ERM) in most real-world applications. Thus the proposed method is a bit impractical.

- About the experiment

The datasets used in the experiments are too small. I suggest authors to perform experiments on larger datasets with high dimensional data points. In addition, I think it would be better to perform experiments on real-world datasets rather than synthetic datasets.

- About writing

The writing needs refinement. The main paper begins to propose the method on page 7, dedicating only four pages to detailing its advantages. I suggest the authors consider moving some background information to the appendix, allowing for a more comprehensive presentation of the proposed method in the main body of the paper.

minors:
The papers should provide the reference of Thm 1.

**Questions:**

see weakness

---

> ### Author Response · Authors · 2024-11-23
> **Authors' response to the Reviewer YmSS**
>
> We acknowledge the reviewer's concerns regarding the limitations of our experimental setup, including the use of a small dataset, the absence of real-world datasets, and the challenges in obtaining the exact $x^*$. These are valid points, and we recognize them as areas for improvement in future work.
>
> However, we would like to emphasize that **the primary focus and contribution of our research lie in the development of a novel "Learning to Optimize" (L2O) ADMM-type framework specifically designed to handle multiblock constrained optimization problems**. These problems are inherently challenging and represent an area of L2O that remains underexplored in the current literature.
>
> Our work lays the methodological foundation for addressing these issues, which we believe is a **critical first step before scaling the approach to larger and more complex datasets or applying it directly to real-world scenarios**. Furthermore, while the current experiments are designed to validate the feasibility and effectiveness of the proposed framework in a controlled setting, we argue that they still provide meaningful insights. This approach allows us to isolate and analyze the core contributions of our method, providing a solid basis for future extensions. We want to mention that in the experiment on optimal transport, we indeed used the **MNIST image data set**, which we recognize as a real-world dataset that is commonly used in the ML community.
>
> We also appreciate your suggestions to improve the structure and clarity of our manuscript. To address this, we will consider **relocating some of the background material to the appendix**. We will also ensure that Theorem 1 is **properly referenced** in the revised version of the paper.

---

> > ### Comment · Reviewer_YmSS · 2024-11-25
> >
> > Thank you very much for your response. After reading your feedback and other reviews, I retain my original score due to the absence of strong theoretical guarantees and the potential for high computational costs.

---

> > > ### Author Response · Authors · 2024-11-25
> > >
> > > Thank you for your prompt reply.
> > >
> > > We acknowledge that the lack of strong theoretical guarantees is a significant limitation of the paper. We understand and respect the possibility of a lower score if the machine learning community prioritizes papers with strong theoretical foundations. However, we would like to emphasize that the computational cost of obtaining labels (i.e., true solutions) in a supervised learning framework is, to the best of our knowledge, an inherent and necessary aspect of learning-to-optimize (L2O) approaches.
> > >
> > > In particular, for L2O, most existing works either assume that the true solution is readily available (e.g., sampled from a known distribution) or rely on running a baseline solver for a significant number of iterations to generate reference solutions. For example, in the context of solving the Lasso problem with L2O approaches, it is common practice to use FISTA (as pointed out by one of the reviewers) for many iterations to obtain accurate reference solutions.
> > >
> > > Moreover, it is worth noting that computational costs associated with data preparation are not unique to our work. Many machine learning frameworks, including large language models (LLMs), require substantial datasets, often at significant cost in terms of privacy, safety, and alignment considerations. These factors highlight the broader challenge of balancing computational costs with the potential benefits of data-driven methods across the ML field.

---

### Note · Authors · 2025-01-25

I have read and agree with the venue's withdrawal policy on behalf of myself and my co-authors.